# Childhood Diarrhea Prevalence and Uptake of Oral Rehydration Solution and Zinc Treatment in Nigeria

**DOI:** 10.3390/children9111722

**Published:** 2022-11-09

**Authors:** Bolaji Emmanuel Egbewale, Omar Karlsson, Christopher Robert Sudfeld

**Affiliations:** 1Takemi Program in International Health, Department of Global Health and Population, Harvard T. H. Chan School of Public Health, Boston, MA 02115, USA; 2Department of Community Medicine, College of Health Sciences Ladoke Akintola University of Technology, Ogbomoso 210214, Nigeria; 3Department of Global Health and Population, Harvard T. H. Chan School of Public Health, Boston, MA 02115, USA

**Keywords:** childhood, diarrhea, death, ORS, zinc, Nigeria

## Abstract

Given the disproportionate burden of childhood diarrhea deaths in Nigeria, this study assessed the prevalence and predictors of the disease and the uptake of ORS and zinc supplementation as treatments in a population-based national survey. Cross-sectional data from the 2018 Nigeria Demographic Health Survey were used. A log-Poisson regression was used to estimate the relative risks (RR) for the individual-level predictors of childhood diarrhea and the uptake of ORS and zinc treatments. A total of 30,713 children under 5 years of age were included in the survey. The period prevalence of reported diarrhea in the last two weeks was 12.9% (95% CI: 12.5%, 13.3%). Among the children with diarrhea, the proportion who received ORS was 39.7% (95% CI: 38.2%, 41.3%), while 29.1% of them received zinc supplements (95% CI: 27.7%, 30.5%), and 21.8% of them received both the ORS and zinc treatments as recommended. Children under 6 months of age with diarrhea had a significantly lower likelihood of being given ORS or zinc when they were compared to the older children. The institutional delivery of them, maternal employment, and improved water sources were also independent predictors of the uptake of ORS and zinc treatments for diarrhea (*p*-values < 0.05). Interventions to prevent childhood diarrhea and improve the coverage of ORS and zinc treatments may reduce the large burden of childhood diarrhea deaths in Nigeria.

## 1. Introduction

Despite there being remarkable reductions in them during the past two decades, 500,000 deaths attributable to diarrhea continue to occur each year among children that are less than 5 years of age [1,2]. The Sustainable Development Goals (SDGs) call on all countries to reduce their under-five mortality to less than 25 per 1000 live births by 2030 [1]. Nigeria currently has an under-five mortality of 114 deaths per 1000 live births [3], and it is estimated that 16% of the remaining child deaths are attributable to diarrhea [4]. In Nigeria, the diarrheal disease was only second to neonatal disorders among the main causes of childhood death in 2019, while other major causes of child death include lower respiratory infections, malaria, and meningitis [4]. Nigeria has less than 5% of the world’s population of children that are under five years old [5], while approximately one-quarter of the global childhood diarrhea deaths occur in the country (~132,000) [2]. Therefore, Nigeria faces a disproportionately large burden of childhood diarrhea deaths.

Diarrhea results in loss of water and electrolytes (e.g., sodium, chloride, potassium), which—if not replaced—can lead to dehydration and child death. The World Health Organization (WHO) and the Government of Nigeria recommend that childhood diarrhea is treated with an oral rehydration solution (ORS) and a zinc supplementation [6,7]. The ORS and zinc supplementation are simple and effective interventions; ORS is estimated to reduce the risk of diarrhea mortality by 93%, and the zinc supplementation reduces the duration of diarrhea, stool output, and risk of persistent diarrhea [8,9]. 

Given the disproportionate burden of childhood diarrhea deaths in Nigeria and the effectiveness of the ORS and zinc supplementations, we examined the population- and individual-level predictors of diarrhea prevalence and the uptake of ORS and zinc treatments. These population- and individual-level predictors may explain the variation in update of these interventions across Nigeria. Based on a literature review, we examined the child, maternal, and household factors that were hypothesized to be associated with diarrhea incidence or treatment uptake. These results are intended to inform the design of the interventions and public health programs to prevent diarrhea and increase the uptake of ORS and zinc supplementations for the treatment of diarrhea among children that are under 5 years of age. 

## 2. Materials and Methods

### 2.1. Study Design

The study used data from the 2018 Nigeria Demographic Health Survey (NDHS) [10]. The 2018 NDHS is a nationally representative cross-sectional household survey that collected data on demographic and health indicators from all 36 states and the Federal Capital Territory. The NDHS used a stratified two-stage cluster sampling design that was stratified by the states and Federal Capital Territory, which were further separated into the urban and rural areas. A total of 1400 enumeration areas (EAs) were selected with probability proportional to size sampling (PPS), and 30 households were randomly selected in each cluster. All of the women aged 15–49 and a subsample of men aged 15–59 were interviewed for the survey in the selected households. The complete details of the sampling methods are found in the 2018 NDHS report. We obtained the approval of the DHS program to use the de-identified and publicly available dataset for this study. 

### 2.2. Study Population

The analytic samples for this study were: (i) all of the children who were under 5 years of age that participated in the 2018 NDHS (n = 30,713) and (ii) the subsample of children that had diarrhea which had been reported by mothers/caregivers during the two weeks prior to the survey (n = 3956). 

### 2.3. Outcomes

The mothers/caregivers were asked whether their child had at least one diarrhea episode during the two weeks prior to the survey visit. Among those that reported diarrhea, the mothers/caregivers were asked if the child received ORS or zinc supplementation as a treatment. ORS was considered to have been given if the mother/caregiver reported the child was ‘given oral rehydration’ or ‘given recommended home solution’.

There were four outcomes which were examined in this study. The first outcome was a maternal report of diarrhea during the prior two weeks among the children under age five (yes/no). Then, among this group, three additional outcomes were considered: whether an ORS treatment was given (yes/no), whether a zinc treatment was given (yes/no), and if both the ORS and zinc treatments (ORS) were given as recommended (yes/no). 

### 2.4. Explanatory Variables

The explanatory variables that were included in the 2018 NDHS were selected before the analysis based on a literature review of child, maternal, and household characteristics that may be associated with diarrhea incidence or treatment. The characteristics of the child included their: age (<6 months, 6–11, 12–23, 24–35, 36–47, or 48–59 months), sex, birth order (first, second and third, fourth and fifth, or sixth and above), whether the child lived with the mother, and the place of delivery (institutional or non-institutional). The maternal/caregiver characteristics were their: age (<20, 20–34, or 35–49 years), education level (no formal education, primary, secondary, and higher education), marital status (never in a union, married/living with a partner, and widowed/divorced/separated), whether she was working, religion (Catholic, other Christian denomination, Islam, Traditionalist, and Others), and ethnicity. The household characteristics included: the region of residence, whether it was an urban/rural residence, the sex of the household head, the source of the drinking water (improved or unimproved), the toilet facility (improved or unimproved), the presence of water at hand washing place, whether there was soap or detergent at the hand washing place, and which quintile the household was in according to their wealth. The household wealth quintiles were directly coded from the 2018 NDHS and were calculated based on ownership of materials and the household characteristics using a principal component analysis. Improved and unimproved drinking water and toilet facilities were defined using the WHO/UNICEF Joint Monitoring Program (JMP) definitions. 

### 2.5. Statistical Analysis

All of the analyses accounted for the design of the 2018 NDHS including the stratification and cluster sampling. Firstly, the descriptive statistics of children under 5 years old, their mothers, and the households were assessed. Secondly, we calculated the proportion of them that had diarrhea and received the ORS and zinc treatment for diarrhea, nationally, for the six geopolitical zones and for 36 Nigerian States and the Federal Capital Territory. Then, we examined the child, maternal and household predictors of childhood diarrhea in univariable and multivariable log-Poisson regression models to estimate the relative risks. The log-Poisson models were selected due to convergence issues for multivariable models; however, the log-Poisson models provided consistent, but not fully efficient estimates of the relative risk and its confidence intervals [11]. Fourthly, we determined the proportion of children with diarrhea that were reported to have received ORS, zinc, and ORS and zinc treatments, and then, we used the univariable and multivariable log-Poisson regression models to estimate the relative risks for the child, maternal and household characteristics [11]. All of the analyses were conducted in Stata version 12 [12]. 

## 3. Results

A total of 30,713 children under 5 years of age that participated in the 2018 NDHS were included in the analysis. The characteristics of households, mothers, and children are presented in Table 1. The child sex distribution in the overall sample was balanced; 50.6% of them were male and 49.4% of them were female, and only 40.5% of the children were delivered in institutional facilities. The majority (70.3%) of the mothers were 20–35 years of age, and approximately 95% of them were either married or cohabitated with partners as at the time of the survey. 

### 3.1. Childhood Diarrhea

A total of 3956 children had mothers/caregivers who reported diarrhea during the 2 weeks prior to the survey. The characteristics of the subsample of children with diarrhea are also presented in Table 1. In the subsample, a similar sex distribution was observed, 50.9% of them were male and 49.1% of them were female, and only 29.8% of the children with diarrhea were delivered in an institutional facility. 

The national period prevalence of childhood diarrhea among children under 5 years of age was 12.9% (95% CI: 12.5%, 13.3%). The period prevalence varied widely by states, ranging from 0.9% in Ogun to 35% in Gombe (Figure 1). The regional diarrhea prevalence ranged from 5.2% in the southwest to 24.6% in the northeast region (Appendix A and Table A1). The table also shows the prevalence of diarrhea by the child, maternal, and household characteristics with childhood diarrhea.

Table 2 shows the results of the univariable and multivariable models that examined risk factors for diarrhea. In the multivariable models, the risk of diarrhea differed by child age; children aged 6–11 months (RR = 2.02: 95% CI: 1.77, 2.32), 12–23 months (RR = 2.04: 95% CI: 1.80, 2.32) and 24–35 months (RR = 1.39: 95% CI: 1.20, 1.58) had an increased risk of diarrhea when they were compared to children that were 0–6 months of age. In addition, children who had a birth order of six or greater had greater a risk of diarrhea when they were compared to firstborns (*p*-Value: 0.006). In terms of the maternal factors, maternal age, maternal education, and maternal employment status were associated with reported diarrhea (*p*-Values < 0.05). As for the household factors, region of residence and household wealth quintile were strongly associated with the risk of diarrhea. The children in the highest wealth quintile had 65% lower risk of having diarrhea (RR = 0.35: 95% CI: 0.31, 0.40) when they were compared to the poorest wealth quintile. Some religious and ethnic groups also had a greater risk of diarrhea in the multivariable models. There was no relationship between water source or sanitation and the risk of reported diarrhea. 

### 3.2. Uptake of ORS and Zinc Supplements for Treatment of Childhood Diarrhea

Among the children that had diarrhea, 39.7% of them were reported to have received the ORS treatment (95% CI: 38.2%, 41.3%), 29.1% of them received the zinc treatment (95% CI: 27.7%, 30.5%), and 21.8% of them (95% CI: 20.5%, 23.1%) were reported to have received the ORS and zinc treatments as recommended. The distribution of the use of the ORS or zinc and both the ORS and zinc treatments varied by state and region as shown in Appendix A and Table A3, Figure 2 and Table A2, and Figure 3, respectively.

Appendix A shows the percentage of children who received the ORS or zinc and ORS and zinc treatments against the child, maternal, and household characteristics. The univariable and multivariable models that assessed the factors associated with uptake of the ORS or zinc and ORS and zinc treatments are presented in Appendix A and Table 3, respectively. In the multivariable models, the children who were 0–6 months of age had a lower zinc and ORS treatment uptake when they were compared to the categories of children who were 6–35 months of age (*p*-values < 0.05), while the children who were delivered in a facility had higher rate of ORS and zinc treatment uptake (*p*-Values < 0.05). 

In terms of maternal factors, the mother not working and them having a maternal age of over 35 were associated with a lower uptake of the ORS treatment (*p*-values < 0.05), while no maternal factors were identified to be associated with the zinc treatment. As for the household factors, an improved water source was associated with an increased uptake of the ORS or zinc and both the ORS and zinc treatments. Some religions, ethnic groups and regions also had greater uptake of the ORS or zinc treatments after an adjustment for other factors. Compared to other regions, both Northwest and Southwest areas appear to have the highest likelihood of utilization of all of the childhood diarrhea recommended treatment methods. 

## 4. Discussion

We found a national diarrhea 2-week period prevalence of 12.9% among children under 5 years of age in Nigeria. The risk of reported diarrhea was found to vary considerably between the states and regions. It is likely that this variation is due to a combination of population- and individual-level factors. We also identified several child, maternal, and household characteristics associated with reported diarrhea including child age, the birth order, maternal age, maternal education, and maternal employment status, the region of residence, and the household wealth quintile. We also found that the standard treatment for acute diarrhea was low with only 39.7%, 29.1%, and 21.8% of the children with diarrhea reporting to have received the ORS, zinc, and ORS and zinc treatments, respectively. The ORS and zinc treatment uptake varied by state and region, and we identified that several child, maternal, and household factors are associated with the receipt of each intervention, notably, children who were 0–6 months old and children in households with unimproved water sources had a lower uptake of both of the diarrhea treatment interventions.

The childhood diarrhea period prevalence in the 2018 NDHS in this study is slightly greater than the national period prevalence of 10.0%, which was reported in the 2013 NDHS [10]. This may be due to there being a minimal difference in the uptake of prevention and diarrhea treatment interventions over time. As a result, there is a clear indication of an urgent need to strengthen the existing national control and preventive efforts against childhood diarrhea as well as to identify new initiatives and programs to reduce the incidence of the diarrhea. Nevertheless, it is important to note that the prevalence of childhood diarrhea in our study is lower when it is compared to those which are seen in other countries in sub-Saharan Africa: Ghana (17%) [13], Kenya (15%) [14], and Uganda (20%) [15].

We found that the child, maternal, and household factors were associated with reported childhood diarrhea. We found reported childhood diarrhea was more prevalent among the children who were 6–11 and 12–23 months old when they were compared to the children who were under six months old. This finding is consistent with the distribution of diarrhea risk that has been reported in previous studies [13,14]. The low risk of childhood diarrhea that is associated with infants that are under 6 months old, among other reasons, may be largely attributed to the effect of exclusive breastfeeding which is known to reduce the incidence of diarrhea in this age group [16]. Moreover, older children exhibit greater mobility, and therefore, may have increased environmental risks for diarrhea [14]. However, in our study, the diarrhea incidence considerably declined after 35 months, and it was the lowest among the children who were 48–59 months old. Older children may have developed stronger immunity against routine childhood diseases if they were compared to the younger children [17,18,19]. We also found that children with a birth order of six or greater had a higher risk of diarrhea, and this likely reflects that there are a larger number of children in the household which may increase their exposure to diarrhea pathogens as well as there being potential differences in their care practices. We also found that children from young mothers who were under 20 years of age had an increased risk of diarrhea which is in line with previous studies in Uganda and Tanzania [15,20,21]. We did not find a significant association between the source of drinking water, toilet facility, or having water or soap in hand washing places with the diarrhea prevalence. A rational explanation for the observed non-significant association is that its availability may not necessarily translate to its usage. Previous cross-sectional studies have also failed to identify an association between childhood diarrhea and sanitation facilities, water sources, or handwashing facilities [14,22]. 

Our study also observed the sub-optimal utilization of ORS treatments with only one in five children with diarrhea having reported to have received the ORS and zinc treatments as recommended. However, it is important to note we also found considerable variation across the thirty-six states and the Federal capital territory of Abuja. A contributor to the variation may be that there are different diarrhea treatment programs and projects between the regions. For example, a program that included policy revision and partner coordination, market-shaping to improve its availability, provider training and mentoring, and caregiver demand generation was found to significantly increase the ORS and zinc treatment coverage in eight states of Nigeria [23]. Therefore, there is an urgent need for the integration of these programs to promote uptake of these diarrhea treatment interventions across the country. Nevertheless, the utilization of ORS and zinc treatments that were recorded in our study are higher than 18.9 and 14.8%, respectively, than those that were found in Sudan [24], but they are similar to the 30% coverage of the ORS treatment in Ethiopia [25], as well as being slightly higher than the 21.5% coverage of the zinc treatment in the East African countries [26]. As a result, this shows that the zinc and ORS treatment coverage remains a challenge in sub-Saharan Africa, and there are differences between the countries in the region as well as significant variation within the country context as shown in our study.

We found individual-level child, maternal, and household predictors of ORS and zinc treatment coverage. Previous studies in Nigeria had associated this low coverage with a poor knowledge of its benefits and preparation in the case of the ORS treatment [27,28,29]. Our study also found a significant association between the child’s age and the uptake of the ORS and zinc treatments during an episode of diarrhea. The coverage of the ORS and zinc treatments were particularly low for the infants that were 0–6 months of age, a group that may also face a greater risk of diarrhea mortality. We hypothesize this is related to exclusive breastfeeding promotion and potential confusion from mothers and health workers that ORS treatments and vitamins can be given, and they are not considered to be mixed with feeding [30,31]. Our findings are consistent with a study that was conducted in Sudan, which also found an increased likelihood of uptake of ORS treatments in older children [24]. We also found a significant association between the uptake of the ORS, zinc, and ORS plus zinc treatments with having an improved source of drinking water. This emphasizes the importance of the availability of clean water for the preparation of the ORS and oral zinc treatments. Researchers have advocated for the production of commercial 1-L water packages as part of the ORS preparation [28]. We found that the children who were born in an institutional facility were more likely to be given an ORS treatment. We think this is likely a proxy for the access and preference for services in the health care system. A study that was conducted in Ethiopia also reported a significant increase in the likelihood of the uptake of an ORS treatment with prenatal clinic attendance [25]. We found no association between maternal education with the ORS and zinc treatment coverage, which is aligned with studies that were conducted in Sudan and Ethiopia [24,26]. However, we did not find a significant association between the rural/urban residence, which contrasts with a study in Sudan [24], where the children in rural settings were less likely to be given the ORS or zinc treatments when they were compared to the urban children. As a result, we found a diverse set of factors associated with the use of the ORS and zinc treatments, and it will be important to consider the local context in the design of interventions that seek to improve the ORS and zinc treatment coverage. 

Our study has several limitations. Firstly, we relied on the occurrence of maternally reported diarrhea as well as zinc and ORS treatments. As a result, the occurrence of a misclassification is possible, and we hypothesize that the mothers may have overreported the occurrence of the zinc and ORS treatments due to its social desirability so the true coverage of these interventions may in fact be lower. In addition, our study was cross-sectional, and as a result, causality cannot be implied. In addition, we did not examine the relationship of the characteristics of the fathers or adult males in the household with diarrhea prevalence and treatment uptake. Last, our study was conducted in Nigeria, and so, the findings may not be generalizable to other settings in sub-Saharan Africa. 

## 5. Conclusions

The childhood diarrhea prevalence remains high while the ORS and zinc treatment coverages are low in Nigeria with there being a large variation across the 36 states and the Federal capital territory. Based on our analysis, these differences in the coverages are likely due to a combination of population- and individual-level factors. We also found a diverse set of child, maternal, and household factors that are associated with diarrhea and the ORS and zinc treatment coverages. Programs should consider these multiple determinants to design interventions. Interventions are urgently needed to reduce the diarrhea incidence, improve the ORS and zinc treatment coverage to reduce the number of diarrhea deaths, and support Nigeria to achieve the Sustainable Development Goals on child mortality by 2030.

## Figures and Tables

**Figure 1 children-09-01722-f001:**
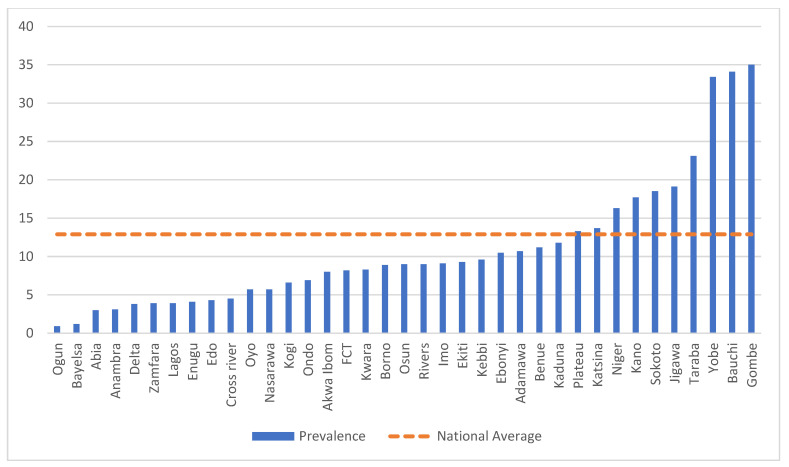
Period prevalence of diarrhea during the two weeks prior to the survey among children under 5 years of age by state compared to the national average of 12.9%.

**Figure 2 children-09-01722-f002:**
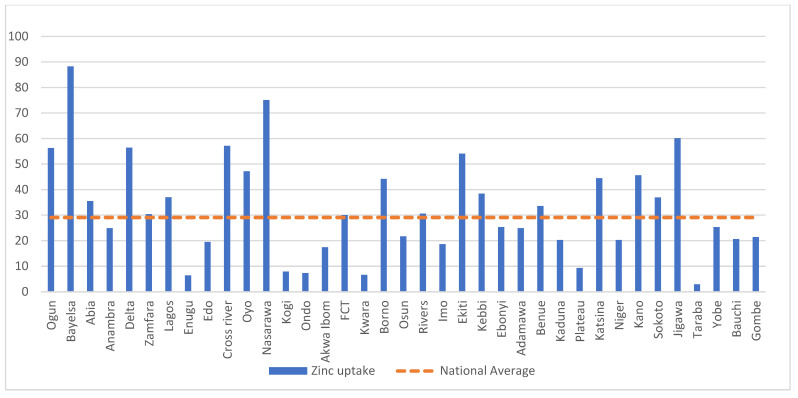
Percentage of children with diarrhea during the 2 weeks prior to the survey that were reported to have received zinc supplements by state compared to the national average of 29.1%.

**Figure 3 children-09-01722-f003:**
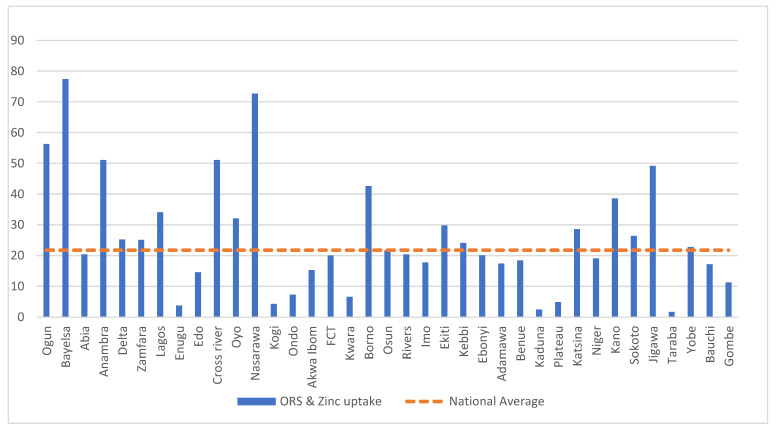
Percentage of children with diarrhea during the 2 weeks prior to the survey that were reported to have received oral rehydration solution (ORS) and zinc supplements by state compared to the national average of 21.8%.

**Table 1 children-09-01722-t001:** Child, maternal, and household characteristics of children under 5 years of age included in the 2018 Nigeria DHS (N = 30,713) and characteristics of the sub-sample of children under 5 years of age for whom mothers/caregivers reported diarrhea during the 2 weeks prior the survey (N = 3956).

	Children under 5 Years of Age	Children under 5 Years of Age with Reported Diarrhea in the Last Two Weeks
	n (%)	n (%)
Child Characteristics		
Child age (months)		
<6	3250 (10.58)	324 (8.19)
6–11	3149 (10.25)	641 (16.20)
12–23	6059 (19.73)	1229 (31.07)
24–35	5834 (19.00)	800 (20.22)
36–47	6168 (20.08)	560 (14.16)
48–59	6253 (20.36)	402 (10.16)
Child sex		
Male	15,537 (50.59)	2015 (50.94)
Female	15,176 (49.41)	1941 (49.06)
Birth order		
First	5885 (16.16)	684 (17.29)
Second/Third	10,504 (34.20)	1228 (31.04)
Fourth/Fifth	7202 (23.45)	883 (22.32)
Sixth and above	7122 (23.19)	1161 (29.35)
Place of delivery		
Non-institutional	18,277 (59.51)	2776 (70.71)
Institutional	12,436 (40.49)	1180 (29.83)
Maternal characteristics		
Maternal age (years)		
<20	1289 (4.19)	246 (6.22)
20–34	21,583 (70.27)	2805 (70.90)
35–49	7844 (25.54)	905 (22.88)
Current marital status		
Never in union	645 (2.10)	59 (1.49)
Married/living with partner	29,181 (95.01)	3799 (96.03)
Widowed/Divorced/Separated	887 (2.89)	98 (2.48)
Maternal education		
No education	13,527 (44.04)	2248 (56.83)
Primary	4776 (15.55)	614 (15.52)
Secondary	9913 (32.28)	931 (23.53)
Higher	2497 (8.13)	163 (4.12)
Mother currently working		
No	9949 (32.39)	1353 (34.20)
Yes	20,764 (67.61)	2603 (65.80)
Household characteristics		
Place of residence		
Urban	10,851 (35.33)	1098 (27.76)
Rural	19,862 (64.67)	2858 (72.24)
Religion of mother		
Catholic	2744 (8.93)	215 (5.43)
Other Christian	9594 (31.24)	722 (18.25)
Islam	18,113 (58.98)	3009 (76.06)
Traditionalist	108 (0.35)	8 (0.20)
Other	154 (0.50)	2 (0.05)
Ethnicity of mother		
Hausa	9576 (31.18)	1511 (38.20)
Fulani	2884 (9.39)	675 (17.06)
Ekoi	144 (0.47)	10 (0.25)
Ibibio	440 (1.43)	32 (0.81)
Igala	288 (0.94)	17 (0.4 3)
Igbo	4200 (13.67)	276 (6.98)
Ijaw/Izion	750 (2.44)	19 (0.48)
Kanuri/Beriberi	746 (2.43)	113 (2.86)
Tiv	723 (2.35)	68 (1.72)
Yoruba	3037 (9.89)	186 (4.70)
Other	7925 (25.80)	1048 (26.52)
Source of drinking water		
Unimproved	12,058 (39.26)	1814 (45.85)
Improved	18,655 (60.74)	2142 (54.15)
Toilet facility		
Unimproved	15,644 (50.94)	2162 (54.65)
Improved	15,069 (49.06)	1794 (45.35)
Presence of water at hand washing place		
Water not available	9577 (39.41)	1350 (44.44)
Water available	14,727 (60.59)	1688 (55.56)
Missing	6409 (20.87)	
Soap or detergent present at handwashing place		
No	16,658 (68.54)	2348 (77.29)
Yes	7646 (31.46)	690 (22.71)
Missing	6409 (20.87)	
Sex of household head		
Male	27,776 (90.44)	3685 (93.15)
Female	2937 (9.56)	271 (6.85)
Household wealth quintile		
Poorest	7081 (23.06)	1303 (32.94)
Poorer	6839 (22.27)	1031 (26.06)
Middle	6509 (21.19)	776 (19.62)
Richer	5747 (18.71)	552 (13.95)
Richest	4537 (14.77)	294 (7.43)
Region		
North Central	5403 (17.59)	554 (14.00)
Northeast	6481 (21.10)	1580 (39.94)
Northwest	8934 (29.09)	1234 (31.19)
Southeast	3545 (11.54)	232 (5.86)
South	3021 (9.84)	162 (4.10)
Southwest	3329 (10.84)	194 (4.90)

**Table 2 children-09-01722-t002:** Factors associated with childhood diarrhea during the two weeks prior to the survey in the 2018 Nigeria DHS (N = 30,713).

	N	Univariable Relative Risk (95% CI)	*p*-Value	Multivariable Relative Risk (95% CI)	*p*-Value
Child characteristics					
Child’s age (months)					
<6	3250	Reference		Reference	
6–11	3149	2.04 (1.8, 2.31)	<0.001	2.02 (1.77, 2.32)	<0.001
12–23	6059	2.03 (1.81, 2.28)	<0.001	2.04 (1.80, 2.32)	<0.001
24–35	5834	1.38 (1.22, 1.55)	<0.001	1.39 (1.20, 1.58)	<0.001
36–47	6168	0.91 (0.80, 1.04)	0.159	0.90 (0.78, 1.05)	0.174
48–59	6253	0.64 (0.56, 0.74)	<0.001	0.69 (0.59, 0.81)	<0.001
Sex of child					
Male	15,537	Reference		Reference	
Female	15,176	0.99 (0.93, 1.04)	0.639	0.97 (0.91, 1.04)	0.428
Birth order					
First	5885	Reference		Reference	
Second/Third	10,504	1.01 (0.92, 1.10)	0.896	1.01 (0.91, 1.12)	0.856
Fourth/Fifth	7202	1.05 (0.96, 1.16)	0.264	0.98 (0.88, 1.12)	0.795
Sixth and above	7122	1.40 (1.28,1.53)	<0.001	1.19 (1.05, 1.35)	0.006
Place of Delivery					
Non-institutional	18,277	Reference		Reference	
Institutional	12,436	0.62 (0.59, 0.67)	<0.001	0.93 (0.85, 1.01)	0.097
Maternal characteristics					
Maternal age (in years)					
<20	1289	Reference		Reference	
20–34	21,583	0.68 (0.60, 0.76)	<0.001	0.86 (0.75, 1.00)	0.055
35–49	7844	0.60 (0.53, 0.68)	<0.001	0.76 (0.64, 0.91)	0.002
Current marital status					
Never in union	645	Reference		Reference	
Married/living with partner	29,181	1.42 (1.11, 1.82)	0.005	0.97 (0.89, 0.07)	0.05
Widowed/divorced/separated	887	1.20 (0.89, 1.64)	0.227	1.01 (0.89, 0.16)	0.29
Maternal education					
No education	13,527	Reference		Reference	
Primary	4776	0.77 (0.71, 0.84)	<0.001	1.16 (1.05, 1.28)	0.002
Secondary	9913	0.56 (0.52, 0.61)	<0.001	1.16 (1.04, 1.30)	0.006
Higher	2497	0.39 (0.34, 0.46)	<0.001	0.97 (0.79, 1.18)	0.756
Mother currently working					
No	9949	Reference		Reference	
Yes	20,764	0.92 (0.87, 0.98)	0.009	1.25 (1.17, 1.35)	<0.001
Household characteristics					
Place of residence					
Urban	10,851	Reference		Reference	
Rural	19,862	1.42 (1.33, 1.52)	<0.001	0.91 (0.84, 0.99)	0.043
Religion					
Catholic	2744	Reference		Reference	
Other Christian	9594	0.96 (0.82, 1.11)	0.589	0.90 (0.76, 1.08)	0.265
Islam	18,113	2.12 (1.86, 2.42)	<0.001	1.21 (0.99, 1.47)	0.067
Traditionalist	108	0.94 (0.48, 1.86)	0.871	1.15 (0.59, 2.24)	0.682
Other	154	0.17 (0.04, 0.66)	0.011	0.39 (0.09, 1.64)	0.199
Ethnicity					
Hausa	9576	Reference		Reference	
Fulani	2884	1.48 (1.37, 1.61)	<0.001	1.11 (0.99, 1.23)	0.053
Ekoi	144	0.44 (0.24, 0.80)	0.007	0.99 (0.52, 1.90)	0.981
Ibibio	440	0.46 (0.33, 0.66)	<0.001	1.08 (0.66, 1.77)	0.755
Igala	288	0.37 (0.24, 0.59)	<0.001	0.60 (0.35, 1.02)	0.057
Igbo	4200	0.42 (0.37, 0.47)	<0.001	0.88 (0.64, 1.22)	0.429
Ijaw/Izon	750	0.16 (0.10, 0.25)	<0.001	0.31 (0.16, 0.57)	<0.001
Kanuri/beriberi	746	0.96 (0.81, 1.14)	0.649	0.74 (0.60, 0.91)	0.004
Tiv	723	0.60 (0.47, 0.75)	<0.001	0.88 (0.66, 1.17)	0.375
Yoruba	3037	0.39 (0.33, 0.50)	<0.001	0.75 (0.60, 0.95)	0.018
Other	7897	0.84 (0.78, 0.90)	<0.001	0.86 (0.77, 0.96)	0.005
Do not know	28	0.23 (0.03, 1.55)	0.13	0.60 (0.09, 3.99)	0.596
Sources of drinking water					
Unimproved	12,058	Reference		Reference	
Improved	18,655	0.76 (0.72, 0.81)	<0.001	1.06 (0.98, 1.13)	0.132
Toilet facility					
Unimproved	15,644	Reference		Reference	
Improved	15,069	0.86 (0.81, 0.91)	<0.001	0.94 (0.86, 1.02)	0.109
Presence of water at hand washing place					
Water not available	9577	Reference		Reference	
Water available	14,727	0.81 (0.76, 0.87)	<0.001	1.08 (1.01, 1.16)	0.032
Missing	6409	1.02 (0.93, 1.10)	0.708	1.04 (0.96, 1.12)	0.316
Soap or detergent present at handwashing place					
No	16,658	Reference		Reference	
Yes	7646	0.64 (0.59, 0.69)	<0.001	0.97 (0.88, 1.06)	0.52
Missing	6409	1.02 (0.94, 1.10)	0.68	-	-
Sex of household head					
Male	27,776	Reference		Reference	
Female	2937	0.70 (0.61, 0.78)	<0.001	1.11 (0.96, 1.28)	0.128
Wealth quintile					
Poorest	7081	Reference		Reference	
Poorer	6839	0.82 (0.76, 0.88)	<0.001	0.94 (0.86, 1.03)	0.214
Middle	6509	0.65 (0.60, 0.70)	<0.001	0.87 (0.78, 0.97)	0.015
Richer	5747	0.52 (0.47, 0.52)	<0.001	0.82 (0.71, 0.94)	0.006
Richest	4537	0.35 (0.31, 0.40)	<0.001	0.68 (0.56, 0.83)	<0.001
Region					
North Central	5403	Reference		Reference	
Northeast	6481	2.38 (2.17, 2.60)	<0.001	2.06 (1.83, 2.31)	<0.001
Northwest	8934	1.35 (1.23, 1.48)	<0.001	1.02 (0.89, 1.16)	0.789
Southeast	3545	0.63 (0.55, 0.74)	<0.001	0.68 (0.49, 0.94)	0.02
South	3021	0.52 (0.44, 0.62)	0.97	0.65 (0.50, 0.85)	0.002
Southwest	3329	0.57 (0.48, 0.66)	<0.001	0.62 (0.50, 0.78)	<0.001

**Table 3 children-09-01722-t003:** Multivariable relative risks for reported receipt of oral rehydration solution (ORS), zinc and ORS and zinc treatment for children with diarrhea during the 2 weeks prior to the survey in the 2018 Nigeria DHS (N = 3956).

Characteristics	N	ORS		Zinc		ORS and Zinc	
		Multivariable Relative Risk (95% CI)	*p*-Value	Multivariable Relative Risk (95% CI)	*p*-Value	Multivariable Relative Risk (95% CI)	*p*-Value
Child’s age in Months							
<6	324	Reference		Reference		Reference	
6–11	641	1.34 (1.11, 1.62)	0.002	1.56 (1.21, 2.02)	0.001	1.53 (1.12, 2.09)	0.006
12–23	1229	1.48 (1.24, 1.76)	<0.001	1.73 (1.36, 2.21)	<0.001	1.73 (1.30, 2.31)	<0.001
24–35	800	1.36 (1.13, 1.63)	<0.001	1.61 (1.25, 2.07)	<0.001	1.55 (1.15, 2.09)	0.004
36–47	560	1.30 (1.07, 1.58)	0.009	1.54 (1.21, 2.04)	0.001	1.49 (1.09, 2.04)	0.014
48–59	402	1.29 (1.05, 1.59)	0.016	1.58 (1.20, 2.08)	0.001	1.56 (1.12, 2.16)	0.008
Sex of child							
Male	2015	Reference		Reference		Reference	
Female	1941	1.02 (0.95, 1.62)	0.565	1.01 (0.92, 1.11)	0.832	1.12 (1.00, 1.26)	0.052
Birth order							
First	684	Reference		Reference		Reference	
Second/Third	1228	1.06 (0.94, 1.20)	0.934	1.02 (0.87, 1.20)	0.798	1.01 (0.83, 1.23)	0.918
Fourth/Fifth	883	1.01 (0.88,1.16)	0.884	1.12 (0.94, 1.33)	0.195	1.07 (0.88, 1.33)	0.474
Sixth and above	1161	1.07 (0.92, 1.23)	0.918	1.05 (0.87, 1.27)	0.585	0.99 (0.79, 1.25)	0.934
Place of Delivery							
Non-institutional	2776	Reference		Reference		Reference	
Institutional	1180	1.25 (1.14, 1.37)	<0.001	1.23 (1.11, 1.39)	<0.001	1.26 (1.09, 1.44)	0.001
Maternal age (in years)							
<20	246	Reference		Reference		Reference	
20–34	2805	1.10 (0.90, 1.35)	0.338	1.19 (0.92, 1.54)	0.189	1.34 (0.96, 1.86)	0.084
35–49	905	0.97 (0.77, 1.23)	0.8	1.14 (0.85, 1.53)	0.383	1.24 (0.86, 1.80)	0.254
Current marital status							
Never in union	59	Reference		Reference		Reference	
Married/living with partner	3799	1.13 (0.76, 1.66)	0.551	1.26 (0.74, 2.14)	0.402	1.39 (0.67, 2.76)	0.398
Widowed/Divorced/Separated	98	1.12 (0.71, 1.75)	0.626	1.27 (0.69, 2.34)	0.444	1.29 (0.58,2.86)	0.536
Maternal educational level							
No education	2248	Reference		Reference		Reference	
Primary	614	0.98 (0.88, 1.10)	0.772	0.95 (0.82, 1.11)	0.517	0.93 (0.78, 1.12)	0.458
Secondary	931	0.88 (0.77, 0.99)	0.039	1.05 (0.90, 1.22)	0.538	0.92 (0.76, 1.11)	0.397
Higher	163	0.93 (0.78, 1.12)	0.466	1.14 (0.90, 1.43)	0.282	1.11 (0.83, 1.49)	0.478
Mother currently working							
No	1353	Reference		Reference		Reference	
Yes	2603	1.17 (1.08, 1.28)	<0.001	1.11 (1.00, 1.24)	0.055	1.28 (1.12, 1.47)	<0.001
Place of residence							
Urban	1098	Reference		Reference		Reference	
Rural	2858	1.00 (0.91, 1.11)	0.07	0.99 (0.88, 1.12)	0.924	0.95 (0.82, 1.11)	0.51
Religion							
Catholic	215	Reference		Reference		Reference	
Other Christian	722	1.03 (0.86, 1.24)	0.716	0.99 (0.76, 1.29)	0.92	0.96 (0.69, 1.33)	0.785
Islam	3009	1.29 (1.03, 1.61)	0.026	1.28 (0.95, 1.74)	0.11	1.81 (1.23, 2.67)	0.003
Traditionalist	8	0.80 (0.22, 2.88)	0.737	0.66 (0.09, 4.57)	0.672	0.99 (0.14, 6.72)	0.988
Other	2	0.85 (0.38, 1.93)	0.7	1.81 (0.63, 5.19)	0.272	1.22 (0.61, 2.41)	0.574
Ethnicity							
Hausa	1511	Reference		Reference		Reference	
Fulani	675	0.71 (0.61, 0.82)	<0.001	0.82 (0.67, 0.95)	0.012	0.66 (0.53, 0.82)	<0.001
Ekoi	10	1.63 (0.93, 2.87)	0.086	2.64 (1.35, 5.20)	0.005	3.28 (1.45, 7.43)	0.004
Ibibio	32	0.66 (0.38, 1.15)	0.146	0.99 (0.51, 1.92)	0.966	0.89 (0.38, 2.10)	0.785
Igala	17	1.01 (0.64, 1.60)	0.95	0.68 (0.43, 1.97)	0.822	0.95 (0.40, 2.26)	0.9
Igbo	276	1.23 (0.94, 1.62)	0.134	0.92 (0.63, 1.55)	0.973	1.14 (0.64, 2.04)	0.647
Ijaw/Izon	19	1.42 (0.89, 2.28)	0.143	2.39 (1.47, 3.87)	<0.001	2.75 (1.45, 5.23)	0.002
Kanuri/beriberi	113	1.16 (0.94, 1.44)	0.171	0.85 (0.59, 1.23)	0.393	0.86 (0.60, 1.29)	0.453
Tiv	68	1.07 (0.76, 1.51)	0.702	1.09 (0.64,1.85)	0.749	1.09 (0.56, 2.14)	0.801
Yoruba	186	0.67 (0.51, 0.88)	0.004	0.61 (0.42, 0.87)	0.007	0.53 (0.34, 0.82)	0.005
Other	1048	0.92 (0.80, 1.06)	0.241	1.07 (0.89, 1.29)	0.461	1.00 (0.80, 1.23)	0.974
Sources of drinking water							
Unimproved	1814	Reference		Reference		Reference	
Improved	2142	1.30 (1.18, 1.41)	< 0.001	1.23 (1.11, 1.37)	<0.001	1.35 (1.18, 1.55)	<0.001
Sex of household head							
Male	3685	Reference		Reference		Reference	
Female	271	0.96 (0.82, 1.12)	0.625	0.87 (0.70, 1.08)	0.214	0.91 (0.69, 1.17)	0.449
Wealth quintiles							
Poorest	1303	Reference		Reference		Reference	
Poorer	1031	0.99 (0.88, 1.11)	0.859	0.89 (0.77, 1.03)	0.13	0.85 (0.72, 1.02)	0.076
Middle	776	1.12 (0.99, 1.27)	0.076	1.21 (1.04, 1.41)	0.013	1.10 (0.92, 1.34)	0.285
Richer	552	1.18 (1.02, 1.37)	0.026	1.39 (1.16, 1.66)	<0.001	1.23 (0.98, 1.53)	0.071
Richest	294	1.52 (1.28, 1.80)	<0.001	1.43 (1.15, 1.79)	0.002	1.41 (1.07, 1.85)	0.015
Region							
North Central	540	Reference		Reference		Reference	
Northeast	554	0.89 (0.77, 1.03)	0.112	0.95 (0.78, 1.16)	0.633	0.87 (0.69, 1.10)	0.243
Northwest	1580	1.09 (0.93, 1.27)	0.284	1.92 (1.55, 2.36)	<0.001	1.55 (1.20, 1.99)	0.001
Southeast	1234	0.88 (0.67, 1.16)	0.367	1.11 (0.70, 1.78)	0.648	1.23 (0.68, 2.24)	0.492
South	232	1.10 (0.86, 1.41)	0.457	1.21 (0.84, 1.73)	0.303	1.41 (0.89, 2.24)	0.147
Southwest	162	1.34 (1.06, 1.69)	0.016	1.53 (1.12, 2.12)	0.008	1.66 (1.14, 2.43)	0.009

## Data Availability

The data that support the findings of this study are openly available from the DHS program at https://dhsprogram.com/data (accessed on 19 October 2021).

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
