# Peer review of "Childhood Diarrhea Prevalence and Uptake of Oral Rehydration Solution and Zinc Treatment in Nigeria"

_children, 2022, doi:10.3390/children9111722_

Round 1

Reviewer 1 Report

A good article- well-written and succinct. Some comments: 
There is wide variation in results from the different states in Nigeria. This is a major finding but the authors discuss it little. One would think that there are significant differences among the states that could explain the variation. I suggest the authors spend some effort to explain to the reader the possible reasons for it. The wide discrepancies in results makes me wonder about the quality of the survey. Could there have been significant variation in which the surveyors went about their task? Could there have been regional differences in competence between the survey teams in different regions? Etc. In short, can the validity of the data be trusted? 
The authors write that "There was no relationship of water source or sanitation with the risk of reported diarrhea." In the discussion they explain this with "A rational explanation for the observed non-significant association is that availability may not necessarily translate to usage."
However, when I look in the table there is a p-value <0,001for both improved water and improved toilet facility. Am I  reading the table incorrectly? 
It is surprising if there was no association between WASH and risk of diarrhoea - even if there is a reference to a study that had similar findings. However, there are many other studies that have found the opposite. The introduction of improved water and sanitation is generally seen as one of the main preventive interventions against diarrhoea since the 19th century and on-wards. Please review the data and writing on this WASH matter.  
The authors write: "...a program that included policy revision and partner coordination, market-shaping to improve availability, provider training and mentoring, and caregiver demand generation was found to significantly increase ORS and zinc treatment coverage in 8 states of Nigeria [23]. 

Were those regions also recorded as having high ORS and zinc use rates in the DHS data? If not, that needs to discussed in relation to ref 23. 

In the conclusion section, the authors write: "Childhood diarrhea prevalence remains high while ORS and zinc treatment coverage is low." What is "high" and "low"? There is no standard to relate these assessments to, nor explanations in the article for what "high" diarrhoea prevalence is and "low" ORS and zinc treatment coverage is. 

Reviewer 2 Report

This manuscript takes an important look at the uptake of simple interventions to decrease diarrhoeal burdens in children < 5 years in Nigeria. Based on their analysis, the authors have shown that uptake is low, providing a base for the initiation of simple and cost-effective programmes, including maternal/care-giver education, to overcome the deficit. Although the authors clarify that their finding may not be generalisable to the rest of sub-Saharan Africa, these may still provide a learning opportunity for other countries with high childhood diarrhoea burdens.

There are a number of areas where some additional clarity may be warranted, particularly as this may affect interventions going forward.

In the methods, the authors mention a subsample of men were interviewed – were any of these results included in this analysis? Did these data differ from those collected from the women and if so in what manner?

In the explanatory variables (page 2) the authors list the age ranges – these should be corrected as follows: <6 months, 6–11, 12–23, 24–35, 36–47, or 48–59 months (not 49-29 months as currently stated).

It would be interesting to know what size the households were e.g. were households with more children more likely to have children < 5 years presenting with diarrhoea, which may reflect on childcare being left to older siblings? Also, were there other children in the household that had diarrhoea at the same time, as this could skew the findings? For instance, if there are a number of children <5 years all with diarrhoea in a household, one would expect that they may be managed in the same manner by the mother/caregiver, which implies that household use of zinc and ORS in Nigeria may be better than the results imply.

The authors refer to improved and unimproved water and sanitation facilities – what does this mean in the Nigerian context? If the household is home treating the water (filtration, chlorine tablets, boiling) was this recorded as improved?

In the results, the rows in some of the tables are misaligned, which made the data difficult to review. I also found the naming of the tables and figures confusing as the first table is S1 and the next S2, although it appears in the text, rather than the end of the document. Are these supplementary tables?

In figure S1, I suggest putting the national average in as a straight line, rather than an additional bar in the bar chart.

Figure S2 and S3 should be combined into a single figure to enable better comparison between zinc, and ORS and zinc supplements. Ideally, these could be overlaid as line graphs on figure 1, so that the reader can more easily relate interventions to diarrhoea rates.

Please also include the region in the figures (e.g. Taraba (NE)) – to make the graphs relatable to the results.

In table S3 (which is misaligned), it appears that there is a trend for females to be less likely to receive both ORS and zinc (p=0.052). Does this reflect a perception of the value of male versus female children? If so, this would be an opportunity for intervention, but the authors have not commented.

The results indicate that certain regions are more likely to be associated with childhood diarrhoea and this is indicated in supplemental figure A1, where certain states appear to have very high rates compared with others. Can these areas be related to conflict zones in Nigeria (https://doi.org/10.1016/S1473-3099(19)30559-6)? This information is important as it may affect the success of any national intervention aimed at decreasing diarrhoeal incidence.

Supplemental figure A1 shows a map of Nigeria with the percentage of diarrhoea in different provinces. It would be helpful if these could be related to the regions mentioned in the text e.g. with bolder lines to indicate the regional borders, to provide more clarity, and if the states could be named so that this can be related to figure S1.
